

# Ancient genetic divergence in bumblebee catfish of the genus *Pseudopimelodus* (Pseudopimelodidae: Siluriformes) from northwestern South America

José D. Rangel-Medrano[1], Armando Ortega-Lara[2,3] and Edna J. Márquez[1]

[1] Facultad de Ciencias, Escuela de Biociencias, Laboratorio de Biología Molecular y Celular, Universidad Nacional de Colombia—Sede Medellín, Medellín, Colombia
[2] Grupo de Investigación en Peces Neotropicales, Fundación para la Investigación y el Desarrollo Sostenible (FUNINDES), Cali, Colombia
[3] Programa de Doctorado en Ciencias-Biología, Facultad de Ciencias, Universidad del Valle, Cali, Colombia

## ABSTRACT

*Pseudopimelodus* is a Neotropical genus of bumblebee catfish, composed of four valid species occurring in both *trans-* and *cis*-Andean rivers of South America. The orogeny of the Andes has led to diversification in the genus *Pseudopimelodus* in Colombia. This study analyzed partial sequences of mitochondrial *cox1* and nuclear *rag2* genes to test the hypothesis that the species, nominally recognized as *P. schultzi* and *P. bufonius* in Colombia, correspond to more than two different evolutionary lineages. Results indicate high levels of genetic divergence among individuals of nominal *P. schultzi* and *P. bufonius,* from *trans-* and *cis*-Andean basins in Colombia. In addition, five divergent lineages of *Pseudopimelodus* were confidently delimited by using a single-locus species-discovery approach and confirmed by species tree analyses. Additionally, molecular-clock dating showed that most diversification processes in *Pseudopimelodus* took place during the Miocene, when Andean tectonic evolution was occurring in northwestern South America. The present study provides, for the first time, phylogeographic insight into this Neotropical genus.

## INTRODUCTION

Pseudopimelodidae is a Neotropical catfish family that includes seven genera and at least 50 valid species, with a distribution ranging from the Atrato River in Colombia to Río de la Plata in Argentina (*Fricke, Eschmeyer & Van der Laan, 2019*). Colombia harbors species belonging to six of the seven genera formally described for this family: *Batrochoglanis* (four species), *Cephalosilurus* (one species) *Cruciglanis* (one species), *Microglanis* (three species), *Pseudopimelodus* (two species), and *Rhyacoglanis* (one species) (*DoNascimiento et al., 2017*). The occurrence of two members of *Pseudopimelodus* has been recorded in different Colombian rivers, but the distribution of both species remains controversial. For example, a checklist of Colombian freshwater fish circumscribed *Pseudopimelodus*

Corresponding author
Edna J. Márquez,
ejmarque@unal.edu.co

*bufonius* (Valenciennes, 1840) to the Caribbean slope rivers (Atrato River, Sinú River, and Catatumbo River), Magdalena-Cauca, Orinoco and Amazon river basins, whereas the endemic species *Pseudopimelodus schultzi* (Dahl, 1955) was endemic to the Caribbean slope rivers and Magdalena-Cauca River basin (*Maldonado-Ocampo, Vari & Usma, 2008*). Conversely, a recent revision excluded *P. bufonius* from the Magdalena-Cauca basin, restricting it to the Caribbean slope rivers and the Orinoco and Amazon river basins (*DoNascimiento et al., 2017*), while *P. schultzi* was restricted to the Caribbean slope rivers, including the type locality (Cereté, Córdoba department) of this species of the Sinú River and the Magdalena-Cauca basin.

Furthermore, no phylogeographic studies of *Pseudopimelodus* have been conducted, leaving a gap in our understanding of the possible effects of geomorphological processes on the diversification of this genus. This gap is important, since phylogeographic analyses may reveal cryptic and deeply divergent lineages that are not reflected in the current taxonomy (*Arbogast & Kenagy, 2001*). Specifically, some geological processes in northwestern South America may have caused vicariant events and subsequent differences in the evolutionary histories of *Pseudopimelodus*: (1) the uplift of central cordillera during the Late Oligocene-Early Miocene, (2) the isolation of the Atrato-Pacific slope and the Magdalena-Cauca basin with the uplift of the Western Cordillera during the Late Oligocene-Middle Miocene (*Duque-Caro, 1990*; *Colletta et al., 1990*; *Kellogg & Vega, 1995*), (3) the isolation of the Llanos basin from the Northern Andean Block by the uplift of the eastern Cordillera during the Middle-Late Miocene (*Hoorn et al., 1995*), and (4) the rise of the Vaupes Arch in the Sub-Andean Foreland region, which divided the present upper Amazonas and upper Orinoco river systems (*Hoorn et al., 1995*; *Lundberg, 1998*). Thus, we hypothesize that the species nominally recognized as *P. schultzi* and *P. bufonius* in Colombia correspond to more than two separate evolutionary lineages, concordant with basin tectonic evolution in northwestern South America.

To test the hypothesis, we analyzed a partial sequence of the mitochondrially encoded cytochrome c oxidase I (*cox1;* COI), which exhibits high rates of molecular evolution, to explore recent divergence events within *Pseudopimelodus*; we then used the nuclear recombination activating gene 2 (*rag2*) to explore more ancient phylogenetic relationships among samples of this genus from northwestern South America. Both genes are commonly used in evolutionary studies, due to their adequate variability and ability to reveal phylogenetic relationships and identify freshwater fish species, including members of the superfamily Pimelodoidea (*Hubert et al., 2008*; *Lundberg, Sullivan & Hardman, 2011*; *Sullivan, Lundberg & Hardman, 2006*). Furthermore, we investigated the putative existence of new species and their historical biogeography by using three species-delimitation approaches and dated species tree analysis. This study offers the first insight into the phylogeographic processes and evolutionary history of pseudopimelodid species from the *cis-* and *trans*-Andean region of Colombia.

## METHODS

### Study area and sampling

This study analyzed a total of 257 Pseudopimelodidae muscle tissues from specimens collected in seven hydrographic sub-zones of Colombia (*IDEAM, 2015*; Table 1, Fig. 1): (1) Magdalena River upper sector, (2) Magdalena River middle sector, (3) Magdalena River and Cauca River- lower sectors and San Jorge River, (4) Cauca River upper and middle sectors, (5) Caribbean drainage (Atrato River and Sinú River), (6) Amazon River hydrographic zone (Orteguaza River and Vaupés River), and (7) Orinoco River hydrographic zone (Negro River and Meta River).

Muscle tissues of specimens from Magdalena River, Cauca River and Atrato River, preserved in 97% ethanol, were provided by Integral S.A., through the scientific cooperation agreement CT-2013-002443. In addition, muscle tissue from *Pseudopimelodus* specimens from Sinú River and Orinoco-Amazon hydrographic zones, as well as members of the genera *Cruciglanis*, *Microglanis* and *Rhyacoglanis*, were collected, identified, and provided through the scientific agreements 00466 FUNINDES-INCODER; 0003, 0033, and 00187 FUNINDES-AUNAP; and 037-2014 FUNINDES-HUMEDALES.

Some collection sites were located in sectors with steep slopes in the Magdalena River-upper sector, upstream of the municipality of Honda, which marks the boundary between the upper and the middle Magdalena. Collection sites in the Cauca River upper and middle sectors were located upstream in the Cauca River canyon, which is the steepest margin of the Antioqueño Plateau in the northern portion of the Central Cordillera (*Restrepo-Moreno et al., 2009*) and marks the boundary between the middle and lower sectors of Cauca River. This landform has been considered a geographic barrier for many fish species (*Dahl, 1971*) and is the focus of the largest hydropower project in Colombia (Hidroituango). Additionally, sampling localities in Magdalena River and Cauca River lower sectors and the San Jorge River correspond to lowlands sites downstream in the Cauca River canyon, including flood plain "ciénagas" that are part of the Momposina depression, one of the most important wetlands in northwestern South America. Finally, collection sites in the *cis*-Andean hydrographic sub-zones of Orinoco (Meta River and Negro River) and Amazon (Orteguaza River and Vaupés River) are separated by the Vaupes Arch, the major drainage that divide the Llanos region of eastern Colombia (*Winemiller & Willis, 2011*).

### DNA extraction, PCR amplification and sequencing

Genomic DNA was extracted using the PureLink® Genomic DNA extraction kit (Invitrogen), following the manufacturer's protocol. For phylogeographic analyses, we amplified a partial region of *cox1* gene in all 257 specimens, using the universal primer cocktail VF2, FishF2, FishR2 (*Ward et al., 2005*), and FR1d (*Ivanova et al., 2007*), as previously published. Furthermore, *rag2* gene was amplified from 77 samples, using the primers MHRAG2-F1 and MHRAG2-R1, as reported by *Hardman & Page (2003)*. For both markers, polymerase chain reactions (PCR) were performed with a total volume of 30 $\mu$l, containing 2.5 $\mu$l of genomic DNA, 3 $\mu$l 10X buffer, 1.5 $\mu$l MgCl$_2$ (50 Mm), 0.75 $\mu$l dNTPs (10 mM), 0.75 $\mu$l of the primer cocktail (0.25 pmol/$\mu$l each), 0.15 $\mu$l of Platinum$^{TM}$ Taq DNA Polymerase (5 U/$\mu$l), and 20.6 $\mu$l of sterile nuclease-free water (Amresco). The thermal

Rangel-Medrano et al. (2020), *PeerJ*, DOI 10.7717/peerj.9028

**Table 1   List of Pseudopimelodidae sequences generated in this study and GenBank sequences used for analysis.**

| Species | Country | Origin | GenBank accession (Haplotype) | | Reference |
|---|---|---|---|---|---|
| | | | *rag2* | *cox1* | |
| *Pseudopimelodus schultzi* | C | HZ1-Magdalena[r] | MH595766 (H18) | MH553588 (H18) | 1 |
| *Pseudopimelodus schultzi* | C | HZ2-Magdalena[r] | MH595752 (H4) | MH553578 (H8) | 1 |
| *Pseudopimelodus schultzi* | C | HZ3-Magdalena[r] | MH595751 (H3) | MH553580 (H10) | 1 |
| *Pseudopimelodus schultzi* | C | HZ3-Cauca[r] | MH595749–MH595750 (H1-2) | MH553573–MH553577 (H3-7), MH553579 (H9), MH553581–MH553582 (H11-H12) | 1 |
| *Pseudopimelodus schultzi* | C | HZ3-San Jorge[r] | MH595768 (H20) | MH553571–MH553572 (H1-2) | 1 |
| *Pseudopimelodus schultzi* | C | HZ4-Cauca[r] | MH595753 (H5), MH595758 (H10), MH595764 (H16), MH595767 (H19) | MH553591 (H21), MH553593–MH553594 (H23-24) | 1 |
| *Pseudopimelodus schultzi* | C | HZ4-Cauca[r] | MH595756 (H8), MH595759–MH595760 (H12-13), MH595763 (H15), MH595765 (H17) | MH553583–MH553586 (H13-16), MH553587 (H17), MH553589–MH553590 (H19-20) | 1 |
| *Pseudopimelodus schultzi* | C | HZ5-Atrato[r] | MH595757 (H9), MH595762 (H14) | MH553592 (H22) | 1 |
| *Pseudopimelodus schultzi* | C | HZ5-Sinú[r] | MH595754–MH595755 (H6-7), MH595759 (H11) | MH553595–MH553596 (H25-26) | 1 |
| *Pseudopimelodus bufonius* | C | HZ6-Orinoco[b] | MH595769–MH595771 (H21-23), MH595773 (H25) | MH553597 (H27), MH553600 (H30) | 1 |
| *Pseudopimelodus bufonius* | C | HZ7-Amazon[b] | MH595772 (H24), MH595774 (H26) | MH553598–MH553599 (H28-29), MH553601–MH553603 (H31-33) | 1 |
| *Pseudopimelodus bufonius* | V | Maracaibo[b] | DQ492359 | — | 2 |
| *Pseudopimelodus mangurus* | B | Paranapanema[r] | — | EU179816 | 3 |
| *Pseudopimelodus mangurus* | B | Uruguay[r] | DQ492360 | — | 2 |
| *Batrochoglanis raninus* | B | Aquarium | — | EU179809 | 3 |
| *Batrochoglanis raninus* | P | Nanay[b] | DQ492361 | — | 4 |
| *Cephalosilurus apurensis* | V | Orinoco[b] | DQ486780 | — | 5 |
| *Cephalosilurus apurensis* | V | Orinoco[b] | — | EU179818 | 3 |
| *Cruciglanis pacifici* | C | Anchicayá[r] | MH595776 (AOL94) | MH553607 (AOL94), MH553608 (AOL23) | 1 |
| *Cruciglanis sp.* | C | Mira[r] | MH595775 (AOL81) | MH553607 (AOL24) | 1 |
| *Lophiosilurus alexandri* | B | São Francisco[r] | — | HM405152 | 6 |
| *Lophiosilurus alexandri* | B | São Francisco[r] | JX899754 | — | 2 |
| *Microglanis sp.* | C | Acacias[r] | MH595777 (AOL95) | MH553604 (AOL95) | 1 |

Rangel-Medrano et al. (2020), *PeerJ*, DOI 10.7717/peerj.9028

**Table 1** (*continued*)

| Species | Country | Origin | GenBank accession (Haplotype) | | Reference |
|---|---|---|---|---|---|
| | | | *rag2* | *cox1* | |
| *Rhyacoglanis anulatus* | C | Negro[r] | (AOL97-98) | MH553605–MH553606 (AOL97-98) | 1 |
| *Rhyacoglanis pulcher* | — | — | — | EU179812 | 3 |
| *Pimelodus yuma* | C | HZ1-Magdalena[r] | MH595780 | MH553610 | 1 |
| *Pseudoplatystoma magdaleniatum* | C | HZ1-Magdalena[r] | MH595781 | MH553611 | 1 |

**Notes.**

C, Colombia; V, Venezuela; B, Brazil; P, Perú; b, Basin; r, River; HZ1, Magdalena river-upper sector; HZ2, Magdalena River-middle sector; HZ3, Magdalena River and Cauca River- lower sectors and San Jorge River; HZ4, Cauca River-upper and middle sectors; HZ5, Caribbean drainage - Atrato River and Sinú River; HZ6, Amazon River basin; HZ7, Orinoco River basin.

[1] This study.

[2] *Sullivan, Muriel-Cunha & Lundberg (2013)*.

[3] C Oliveira, 2018, pers. comm.

[4] *Sullivan, Lundberg & Hardman (2006)*.

[5] *Hardman & Lundberg (2006)*.

[6] *De Carvalho et al. (2011)*.

GPS coordinates for each haplotype are provided in the Supplemental Information.

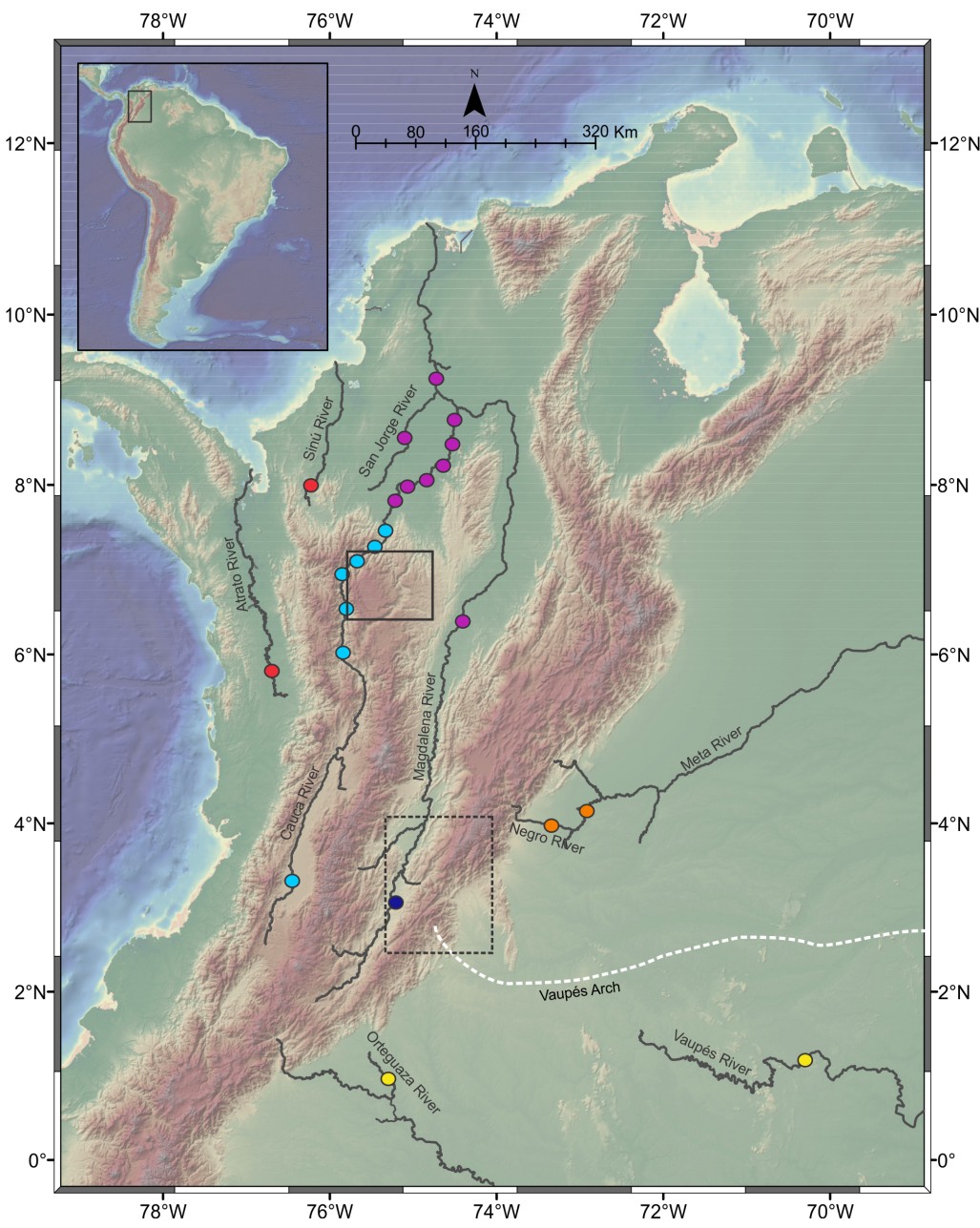

**Figure 1** **Sampling localities of *Pseudopimelodus* species in different *trans-* and *cis-*Andean rivers in Colombia.** Hydrographical sub-zones are denoted by colored circles : Magdalena River-upper sector (dark blue), Magdalena River middle and lower sectors, Cauca River- lower sectors and San Jorge River (purple), Cauca River-upper and middle sectors (light blue), Caribbean drainage - Atrato River and Sinú River (red), Amazon River basin (yellow), Orinoco River basin (orange). Location of Antioquian Plateau in the northern Central Cordillera (solid square), Garzón Massif at the southern tip of the Eastern Cordillera (doted rectangle) and Vaupés Arch (white dashed line) are shown. Map image layer by NOAA National Centers for Environmental Information (NCEI).

profile for *cox1* consisted of an initial cycle at 95 °C for 3 min, 32 cycles of 94 °C for 3 min, 25 s at 60 °C, and 30 s at 72 °C, and the final extension was omitted. A similar thermal profile was used to amplify *rag2*, but with changes in annealing temperature (58 °C) and extension time (45 s). PCR products were confirmed by agarose gel electrophoresis and EZ vision staining and sequenced in both forward and reverse directions using an ABI sequencer 3730XL. Sequences were checked and edited using Geneious v10.0.9 (https://www.geneious.com) software, and multiple alignment to create consensus sequences was performed with the MAFTT v7.308 (*Katoh & Standley, 2013*). Both genes were translated into amino acids to confirm absence of stop codons or unexpected frameshift errors using the sequence translation tool Transeq, available in The European Molecular Biology Open Software Suite (EMBOSS https://www.ebi.ac.uk/Tools/st/emboss_transeq/).

## Genetic diversity and phylogenetic relationships

Calculation of genetic diversity indices, such as number of haplotypes, haplotype (Hd) and nucleotide ($\pi$) diversity (*Nei, 1987*), and polymorphic sites, was performed using DNAsp v6 (*Rozas et al., 2017*). Phylogenetic analysis for each gene and concatenated dataset was conducted using Bayesian Inference, with MrBayes (MB) v3.2 (*Ronquist et al., 2012*), based on the best-fit model estimated in the software IQ-TREE v1.6.12 (*Kalyaanamoorthy et al., 2017*). Parameters included two independent Markov Chain Monte Carlo (MCMC) iterations for 20 million generations sampled every 1000 generations, discarding the first 25% sampled generations as burn-in; settings for remaining parameters were left at their default values. Convergence of the MCMC was based on the Potential Scale Reduction Factor (PSRF) approaching 1.0 and the standard deviation of split frequencies approaching 0.0. In addition to sequences obtained in this study, the phylogenetic analysis included previously published *rag2* and *cox1* sequences of species in all other genera of the family Pseudopimelodidae and the pimelodid species *Pseudoplatystoma magdaleniatum* and *Pimelodus yuma* as outgroups (Table 1). In addition, spatial relationships of *cox1* haplotypes were explored by constructing a haplotype network based on the Median-Joining algorithm (*Bandelt, Forster & Röhl, 1999*), using the software PopART v1.7 (*Leigh & Bryant, 2015*).

## Species delimitation

Mitochondrial *cox1* gene sequences were used to implement three single-locus species-discovery (SLSD) methods in R v3.4.163, by using the scripts developed by *Machado et al. (2018)* and available at https://github.com/legalLab/publications. These methods were implemented to define a priori clusters of putative species (lineages) of *Pseudopimelodus* as trait values for StarBEAST2 v14 (*Ogilvie, Bouckaert & Drummond, 2017*) species tree analysis. The three methods implemented corresponded to (1) a divergence threshold optimizing and clustering approach (locMin; *Brown et al., 2012*), (2) the general mixed Yule coalescent model (GMYC; *Pons et al., 2006*; *Fujisawa & Barraclough, 2013*), and (3) a Bayesian approximation of the GMYC (bGMYC; *Reid & Carstens, 2012*). For all analyses, a set of ultrametric trees was generated with BEAST v1.8.4 (*Drummond & Rambaut, 2007*) choosing HKY+G as the best-fit substitution model, selected by IQ-TREE v1.6.12, setting a relaxed molecular clock model and a Birth-Death Process as a tree prior (*Gernhard,*

*2008*). We ran three independent MCMC chains for 20 million sampled every 18,000 generations, discarding the first 10% of trees as burn-in. Results were combined, and then we sub-sampled a total of 1,000 trees for subsequent analyses. Tracer v1.7 (*Rambaut, 2014b*) was used to verify convergence and ESS values (>300).

All three analyses were based on point estimates from the maximum clade credibility tree generated using TreeAnnotator v1.8.4 and calculation of confidence intervals from the posterior sample of 1,000 trees. The R packages bGMYC v1.0.2 (*Reid & Carstens, 2012*), splits v1.0-19 (*Fujisawa & Barraclough, 2013*), and ape v4.1 (*Paradis, Claude & Strimmer, 2004*) were used in these analyses. A conservative posterior probability of con-specificity at 0.05 were used to summarize the bGMYC posterior samples into putative species.

## Species tree and divergence time estimation

We combined sequence information from both mitochondrial *cox1* and nuclear *rag2* markers to implement a dated species tree considering the molecular substitution model of each gene using StarBEAST2 v14 (*Ogilvie, Bouckaert & Drummond, 2017*) implemented in BEAST2 v2.6.2 (*Bouckaert et al., 2014*). For both markers we set a strict molecular clock, constant population as population size model and Yule Process for species tree prior. To estimate the time to the most recent common ancestor (TMRCA), and the corresponding credibility intervals (95% HPD) for the main lineages, two fossil calibrations points were included: one corresponds to the oldest pimelodid fossil from the Paleogene of South America (*Gayet & Otero, 1999*) that provide a minimum age of 55.8 million years ago (mya), for the divergence between the families Pimelodidae and Pseudopimelodidae. This single calibration age constraint was set in the clade including the two pimelodids species used as outgroups, specifying a normal distribution prior, with a standard deviation equal to 1. Furthermore, an additional fossil calibration corresponding to a cf. *Cephalosilurus* or *Pseudopimelodus* from South America middle Miocene (lognormal distribution, mean 0.1, SD 0.8, offset 11.5, range 15.9–11.5 mya; *Lundberg, 1998*; *Lundberg et al., 2010*) was incorporated on the ancestral node containing all *trans-* and *cis*-Andean *Pseudopimelodus*. Analysis was run for 150,000,000 generations sampling every 15,000 generations and discarding the 10% of the burn-in samples. Convergence was assessed using Tracer v1.7. The summary of all trees and the annotation of mean ages of all nodes, and the corresponding HPD ranges, was conducted with TreeAnnotator v2.6.0. Finally, the selected tree was visualized using the software FigTree v1.4.3 (*Rambaut, 2014a*).

We used the MODEL_SELECTION Package v1.5.2 implemented in BEAST2 v2.6.2 to test alternative species tree models for *Pseudopimelodus* based on the results of SLSD approach. Analyses were performed using the PathSampler icon on the Package Application Launcher, with a path sampling of 10 steps, a chain length of 20 million generations and a burn-in of 10% with no pre burn-in phase. Remaining parameters were left as default following *Leaché et al. (2014)*. Marginal likelihood estimates (MLE) were obtained for each different model run and Bayes Factor Delimitation methods were used to rank the different species delimitation models. Bayes Factors (BF) were calculated by subtracting the MLE among these models, and then multiplying the difference by two [BF = 2 × (MLE1 −

MLE0)]. A positive BF value indicates support in favor of model 1, while a negative BF value indicates support in favor of model 0 (*Leaché & Bouckaert, 2018*).

## RESULTS

### Sequence analysis and phylogenetic relationships

Sequence edition yielded a final alignment matrix of a 357 bp and 520 bp fragment for *cox1* and *rag2,* respectively. No evidence of indels or stop codons was found after translation into amino acids, indicating that the amplified product corresponds to functional *cox1* and *rag2* sequences. For the *cox1* dataset, sequence analyses revealed 33 different haplotypes out of 247 sequences analyzed (GenBank accessions MH553571–MH553603; MH800618–MH800835), defined by 72 polymorphic sites, 52 of which were parsimoniously informative. Values of haplotype and nucleotide diversities for *cox1* dataset were $h = 0.761 \pm 0.020$ and $\pi = 0.046 \pm 0.001$, respectively. Additionally, sequence analysis of *rag2* in the *trans*-Andean *Pseudopimelodus* revealed 26 haplotypes of 77 sequences analyzed (GenBank accessions MH595749–MH595774; MH800567–MH800617), defined by 22 polymorphic sites, 16 of which were parsimoniously informative. Values of haplotype and nucleotide diversities were $Hd = 0.878 \pm 0.024$ and $\pi = 0.010 \pm 0.000$. According to the Bayesian Information Criterion, the optimal model for *cox1* and *rag2* datasets were HKY + G4 and HKY respectively. Collection information, voucher numbers and GenBank accessions for all new sequences obtained in this study are provided in Tables S1 and S2.

All three SLSD methods consistently suggested five unique molecular delimitations corresponding to lineages 1–5 (Fig. 2). However, we found incongruence among the methods for the delimitation of the valid species *P. mangurus* (EU179816) and *P. charus* (EU179815). *Pseudopimelodus mangurus* was only correctly delimited by GMYC, while *P. charus* was correctly delimited by locMin and GMYC (Fig. 2). The point estimates were 13 for bGMYC, 15 for locMin, and 16 for GMYC. Confidence intervals (95%) were largest for locMin (12–16 species, mean: 14), followed by GMYC (5–19 species, mean: 16), and lowest for bGMYC (4–15 species, mean: 11, median: 12, mode: 14).

Median joining network (Fig. 3) confirmed these lineages and also showed a deeper genetic divergence between lineage 1 and the remaining *Pseudopimelodus* lineages, as evidenced by the largest number of mutation steps (19 point mutations). Notably, lineage 1 and lineage 5, represented by a higher number of samples, showed unique haplotypes related to a common ancestral haplotype distributed in different sampling sites (Fig. 3). Network analysis showed a lineage composed of haplotypes from the Magdalena River middle sector, Magdalena River and Cauca River lower sectors and the San Jorge River (lineage 1); haplotypes from Amazon River hydrographic zone (lineage 2) and Orinoco River hydrographic zone (lineage 3); haplotypes from Atrato River and Sinú River (lineage 4); and haplotypes from the Magdalena River upper sector and, Cauca River upper and middle sectors (lineage 5).

The coalescent-based analyses using the MODEL_SELECTION Package ranked the GMYC species delimitation model as the more likely hypothesis (MLE = −4038.037016) instead of the alternatives bGMYC (MLE = −4042.52818) or locMin (MLE =

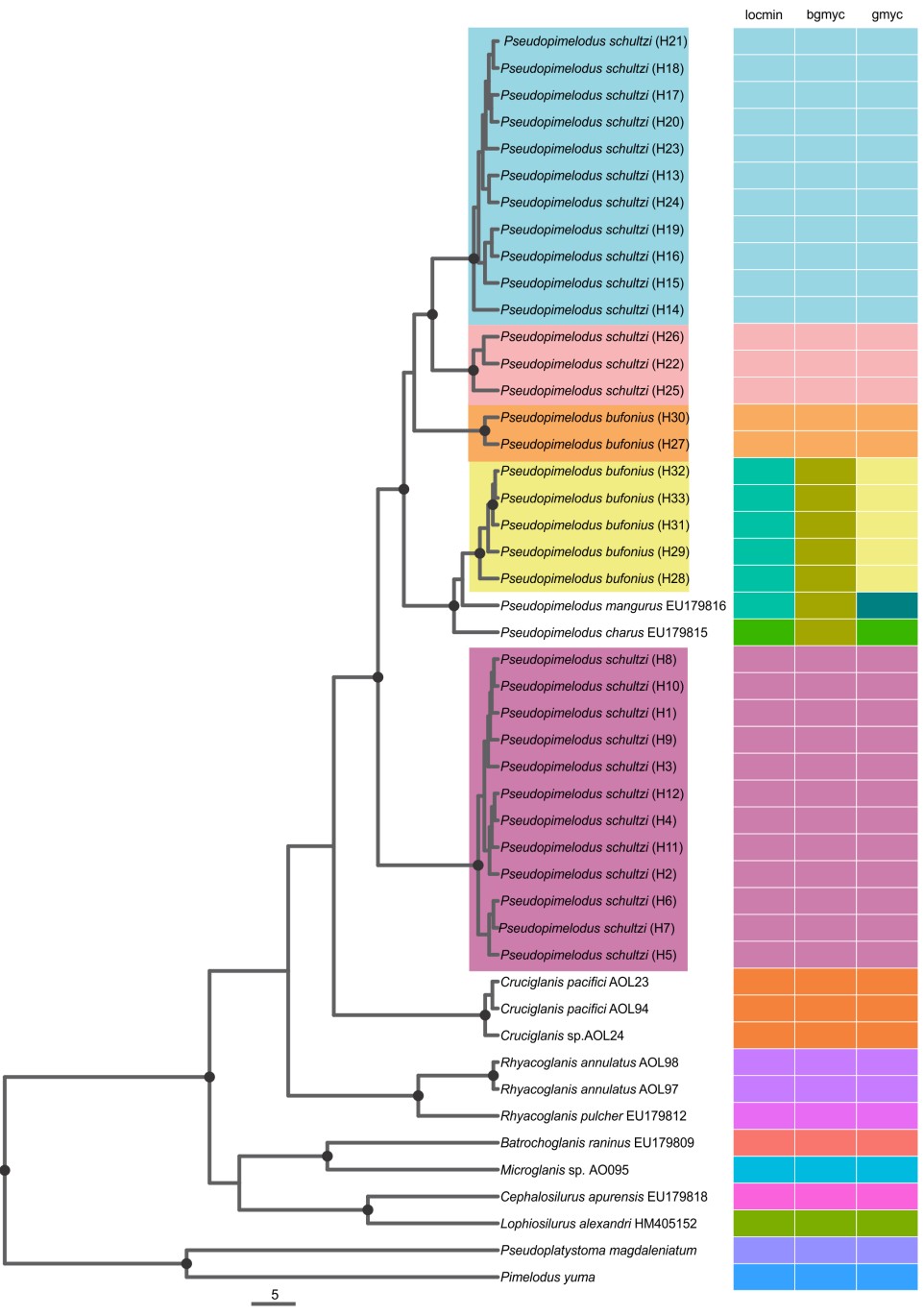

**Figure 2   Single Locus Species Discovery Analysis showing the Maximum clade credibility tree from BEAST.** Bayesian posterior probabilities above 0.95 are shown as dark nodes. Point estimate species delimitations are shown by method as colored boxes. The tree was presented using the ggtree_1.6.11 package (*Yu et al., 2016*).

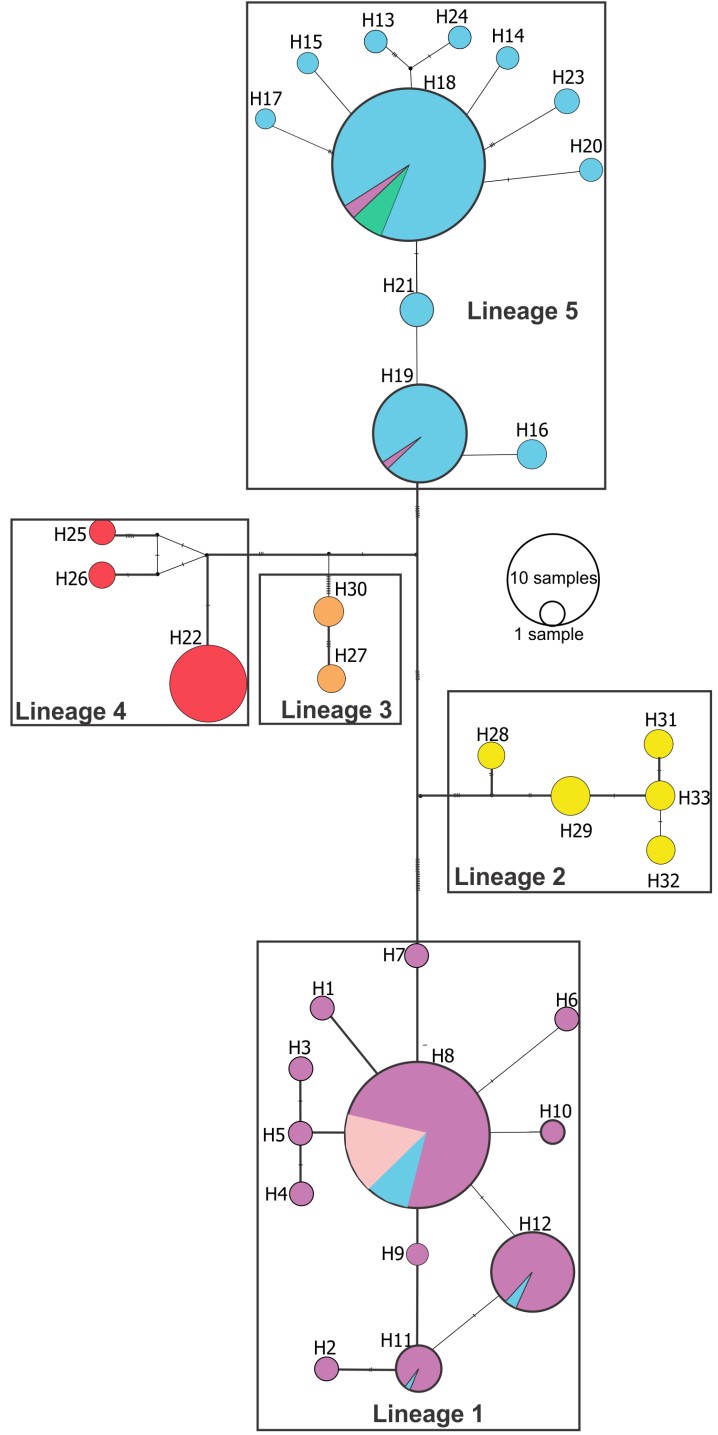

**Figure 3  Median-joining network of *Pseudopimelodus cox1* haplotypes.** Hydrographical sub-zones are denoted by the colors of circles: Magdalena River and Cauca River- lower sectors and San Jorge River (purple), Cauca River-upper and middle sectors (blue), Magdalena River middle sector (pink), Magdalena River-upper sector (green), Caribbean drainage - Atrato River and Sinú River (red), Orinoco River basin (orange) and Amazon River basin (yellow).

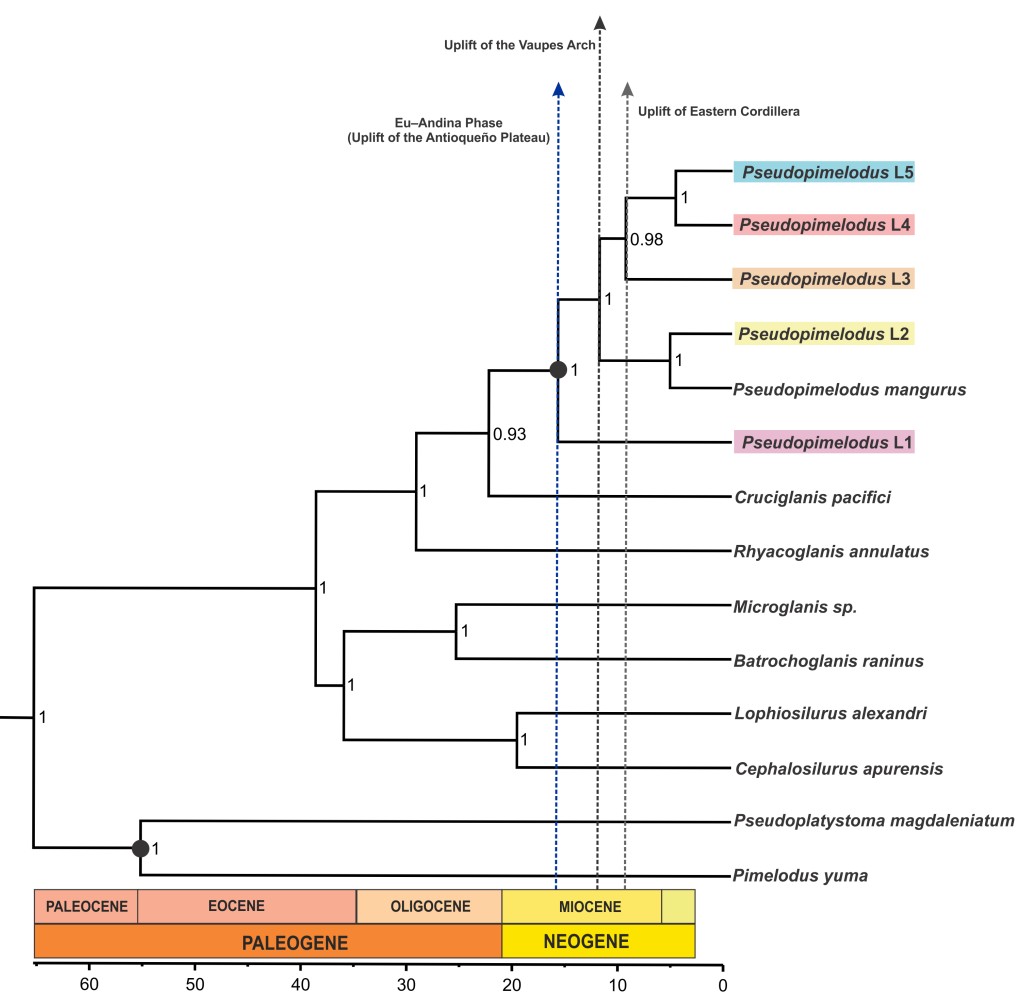

**Figure 4 Dated species tree of the family Pseudopimelodidae generated using StarBEAST2.** Vertical dashed blue, black and grey lines indicates uplift of the Antioqueño Plateau, Vaupés Arch and Eastern Cordillera, respectively. Calibrated nodes are denoted by a black circle; time in mya is indicated by the scale bar.

−4080.28218) models. Moreover, the five evolutionary lineages of *Pseudopimelodus* were also supported by the dated species tree using the concatenated *rag2* and *cox1* genes (Fig. 4). This analysis in conjunction to all three SLSD models and phylogenetic analysis using MrBayes (Supplementary Figs. 1–3) showed that *cox1* and *rag2* recovered the genus *Pseudopimelodus* as a monophyletic group. In addition, SLSD results and dated species tree support the idea that the genus *Rhyacoglanis* is a sister clade of *Pseudopimelodus* plus *Cruciglanis*.

Divergence time estimates (Fig. 4) indicated that lineage 1 diverged from all remaining members of *Pseudopimelodus* during Middle Miocene (16.21 mya 95% HDP 11.91–20.79). Similarly, a Middle Miocene split was detected between lineage 2 and the clade comprising lineages 3, 4 and 5 (12.37 mya HPD 8.98–16.23), while a Late Miocene divergence was

detected between lineage 3 and the clade formed by lineages 4 and 5 (9.93 mya HPD 6.54–13.20), and between lineages 4 and 5 (5.28 mya HPD 1.61–8.20).

## DISCUSSION

This study tested the hypothesis that the species nominally recognized as *P. schultzi* and *P. bufonius* in Colombia correspond to more than two different evolutionary lineages, the divergence times of which agree with Andean tectonic evolution in northwestern South America. Our expectation was that geological events induced vicariance in *Pseudopimelodus* from northwestern South America. Despite the limitations of using short fragments, the use of nuclear and mitochondrial molecular markers, with different mutational rates, allowed us to infer ancient, as well as more recent, genetic divergence in this group of bumblebee catfish. In this study, we did not cover the complete geographical range where nominal *P. bufonius* occurs in Colombia; instead, we analyzed only a small number of individuals from the *cis*-Andean portion. Thus, the detected haplotypes may not represent all genetic lineages present in this geologically complex region.

Mitochondrial *cox1* and nuclear *rag2* analyses supported our hypothesis of divergent lineages of *Pseudopimelodus* inhabiting *trans* and *cis*-Andean rivers in Colombia. Indeed, the molecular evidence gathered in this study supports the non-monophyly of *P. bufonius* and *P. schultzi*, based on the strong genetic divergence among specimens identified as *P. schultzi* representing populations from Magdalena, Cauca, Atrato, and Sinú rivers, and among nominal *P. bufonius* from the Amazon and Orinoco rivers. Specifically, *trans*-Andean *P. schultzi* includes three divergent lineages: (1) Magdalena River lower and middle sectors, Cauca River lower sector and San Jorge River, (2) Magdalena River upper sector, Cauca River upper and middle sectors, and (3) Sinú and Atrato rivers. Moreover, *P. bufonius* populations from the *cis*-Andean region of Colombia include two divergent lineages: (1) Orinoco River hydrographic zone and (2) Amazon River hydrographic zone.

We found perfect congruence among all SLSD methods in delimiting the above-mentioned lineages, and species tree analysis supported these results, indicating that at least four new candidate species, previously synonymized with *P. schultzi* and *P. bufonius*, occur in Colombia. This outcome suggests that a robust taxonomic revision of *Pseudopimelodus*, and especially *P. bufonius*, is needed; future studies should include samples from the type locality (Cayenne, French Guiana; *Boeseman, 1972*), since considering our results, it is reasonable to think that the species present in the Colombian *cis*- and *trans*-Andean rivers may not correspond to *P. bufonius sensu stricto*.

Furthermore, inter-generic relationships detected in this study, especially those observed with *cox1* results using BEAST and SLSD methods that showed that *Pseudopimelodus* is sister to *Cruciglanis,* and this clade is sister to *Rhyacoglanis,* support recent phylogenetic studies on morphological characters and data of gross morphology from the brain of Pseudopimelodidae (*Shibatta & Vari, 2017*; *Abrahão, Pupo & Shibatta, 2018*). However, further research, using more markers and a larger sample size, is required to completely clarify the phylogenetic relationships of the family Pseudopimelodidae and its place in a biogeographic framework of Neotropical diversification (*Machado et al., 2018*).

Given the geographic distribution of each identified lineage, and the general branching of the tree topology, the results agree with the geological history of the Andean mountains in Northwestern South America and support previously proposed models of ichthyofauna division, based on several species-level phylogenies (*Albert, Lovejoy & Crampton, 2006*). Patterns of divergene, in concordance with the geological evolution of Northwestern South America found in this study, have also been reported in other fishes (*Hernández et al., 2015*; *Machado, Galetti & Carnaval, 2018*; *Rincón-Sandoval, Betancur & Maldonado-Ocampo, 2019*) and other taxa such as mammals (*Coimbra et al., 2017*) and insects (*Salgado-Roa et al., 2018*; *Bartoleti et al., 2018*). In addition, our results support a north to south uplift of the Colombian Andes (*Florez, 2003*), as diversification of *Pseudopimelodus* occurred in a sequential manner from older lower altitude diversification events to younger higher altitude diversification events, as was also reported in Neotropical plant species in Colombia (*Richardson et al., 2018*).

The Middle Miocene divergence of lineage 1, which exhibited the deepest divergence of the detected lineages, strongly agrees with the onset of the Eu-Andina phase of the Northern Andes (18 mya to present), previously reported by *Van der Hammen (1960)*. During this phase, the Antioqueño Plateau (the largest high-elevation erosional surface in the Northern Andes) attained its actual elevation in the central Cordillera, from the raising of the erosion surface from close to sea level to 3,600 m a.s.l. (*Restrepo-Moreno et al., 2009*). This result is concordant with the fact that lineage 1 is predominantly present down-stream of the Cauca River canyon, which is the steepest margin of the Antioqueño Plateau (*Restrepo-Moreno et al., 2009*).

Furthermore, the divergence between lineage 2 and the clade containing lineages 3, 4 and 5 was estimated to be at 12.37 mya (HPD 8.98–16.23). This estimated date is consistent with the first development of the Amazon River during the late-middle Miocene, when the paleo-Amazon River was still partly connected to the paleo-Orinoco, through a fluvio-lacustrine system (*Hoorn, 1993*; *Hoorn et al., 1995*). Then, the "paleo-Amazon-Orinoco" split into two different Atlantic-draining basins at approximately 8–10 mya, as result of the elevation of the Vaupes Arch in the region comprising the southern area of the Colombian Llanos (*Cooper et al., 1995*; *Hoorn et al., 1995*; *Lundberg, 1998*). Previous studies have also yielded evidence of allopatric sister lineages of freshwater fish, isolated by this Amazonas-Orinoco vicariance event (*Sivasundar, Bermingham & Orti, 2001*; *Winemiller et al., 2008*; *Machado et al., 2018*).

The Late Miocene divergence (9.93 mya HPD 6.54–13.20) observed between lineage 3 and the clade containing lineages 4 and 5, agrees with the uplift of Eastern Cordillera (10 mya; *Gregory-Wodzicki, 2000*) that permanently divided the former foreland basin into the Magdalena and Llanos basins (*Hoorn et al., 1995*). However, the Eastern Cordillera was not a substantial orographic barrier, as there must have been a low-elevation fluvial corridor between the Magdalena Valley and the eastern Orinoco plains connecting these two geographic provinces (*Richardson et al., 2018*). This may explain the closer phylogenetic relationships detected between lineage 3 (haplotypes from Orinoco hydrographic zone) and the *trans*-Andean lineages 4 and 5. This fluvial corridor may correspond to the area of the Garzón Massif (between latitudes 1°–4°N), located in the southeastern flank of the

Eastern Cordillera in the Colombian Andes (*Altenberger et al., 2012*), which has suffered a more recent and rapid exhumation process (*Anderson et al., 2016*). In fact, *Anderson et al. (2016)* states that an important barrier to dispersal was not fully established in the Eastern Cordillera until the latest Miocene-Pliocene (ca. 6–3 mya) phase of the rapid exhumation of the Garzón basement.

Moreover, the splitting of lineage 4 from lineage 5 occurred during the late Miocene (5.28 mya HPD 1.61–8.20), concordant with the uplift of the Western Cordillera that caused the isolation of the trans-Andean Atrato-Pacific slope from the Magdalena basin during Late Miocene to Pliocene (*Kellogg & Vega, 1995*). In addition, the close relationship of samples from the Atrato and Sinú rivers (lineage 4) supports the previous hypothesis that a historical connection resulting from headwater stream capture existed between the adjacent Atrato and Sinú basins that was posteriorly isolated by the uplift of Western Cordillera during the Miocene (*Rincón-Sandoval, Betancur & Maldonado-Ocampo, 2019*).

Altogether, the results obtained in this study provide a better understanding of the phylogeographic processes that govern the current distribution and diversification of freshwater ichthyofauna in northwestern South America. Moreover, as shown by SLSD and species tree results, the taxonomic diversity within *Pseudopimelodus* is currently underestimated, which can affect the estimation of species richness, jeopardize the understanding of ecological patterns, and may be harmful to species at a high risk of extinction (*Vogel Ely et al., 2017*). In light of these results, ongoing studies of morphological analyses, including osteology and external morphology, we are describing two new *Pseudopimelodus* species from the Magdalena-Cauca River Basin, corresponding to lineages 1 and 5 described herein (Restrepo-Gómez et al., unpublished data, 2020). Finally, and independently of taxonomic implications, genetic divergence of *Pseudopimelodus* lineages must be considered in current conservation and management policies of Colombian bumblebee catfish fisheries.

## CONCLUSIONS

This study provides strong mtDNA evidence that the genus *Pseudopimelodus* in Colombia comprises at least five different, clearly delimited, evolutionary lineages. The lineages exhibit phylogenetic histories that agree with the geological history of northwestern South America. We recommend that future genetic studies include more genetic markers and a greater number of samples, from additional localities, to examine more thoroughly and provide a deeper understanding of the phylogeographic patterns of this genus in South America. These results contribute to the phylogeographic knowledge of Pseudopimelodidae inhabiting *cis* and *trans*-Andean river systems in Colombia and provide an important benchmark for future phylogenetic studies in this diverse and widespread Neotropical catfish family.

## ACKNOWLEDGEMENTS

This paper is dedicated to the memory of our admirable colleague and friend Dr. Javier Alejandro Maldonado Ocampo, who recently passed away in a tragic accident in the Vaupés

River. The authors would also like to acknowledge the valuable comments of Carolina de Barros Machado da Silva and the anonymous reviewers that improved the manuscript.

### Funding

This research was supported by the scientific cooperation agreement between Universidad Nacional de Colombia and Integral S.A., on 19th September 2013 and Universidad Nacional de Colombia, Sede Medellín and Empresas Públicas de Medellín, Grant CT-2013-002443 "Variación genotípica y fenotípica de poblaciones de especies reófilas presentes en el área de influencia del proyecto hidroeléctrico Ituango", Grant CT-2019-000661 "Variabilidad genética de un banco de peces de los sectores medio y bajo del río Cauca". The funders had no role in study design, data collection and analyses, decision to publish, or preparation of the manuscript.

### Grant Disclosures

The following grant information was disclosed by the authors:
The scientific cooperation agreement between Universidad Nacional de Colombia and Integral S.A.
Universidad Nacional de Colombia, Sede Medellín and Empresas Públicas de Medellín: CT-2013-002443.
"Variación genotípica y fenotípica de poblaciones de especies reófilas presentes en el área de influencia del proyecto hidroeléctrico Ituango": CT-2019-000661 "Variabilidad genética de un banco de peces de los sectores medio y bajo del río Cauca".

### Competing Interests

The authors declare there are no competing interests.

### Author Contributions

- José D. Rangel-Medrano conceived and designed the experiments, performed the experiments, analyzed the data, prepared figures and/or tables, authored or reviewed drafts of the paper, and approved the final draft.
- Armando Ortega-Lara analyzed the data, prepared figures and/or tables, authored or reviewed drafts of the paper, and approved the final draft.
- Edna J. Márquez conceived and designed the experiments, analyzed the data, prepared figures and/or tables, authored or reviewed drafts of the paper, and approved the final draft.

### DNA Deposition

The following information was supplied regarding the deposition of DNA sequences:
The sequences are available in GenBank: *cox1* (MH553571 to MH553611) and *rag2* (MH595749–MH595781) sequences of *Pseudopimelodus, Cruciglanis, Microglanis* sp., *Rhyachoglanis*; Outgroups: *Pimelodus yuma* - Pyuma-UNAL001 (rag2: MH595780; *cox1*: MH553610); *Pseudoplatystoma magdaleniatum* Pmag-UNAL002 (*rag2*: MH595781; *cox1*: MH553611).

## Data Availability

The sequences are available in the Supplemental Files.

## Supplemental Information

Supplemental information for this article can be found online at http://dx.doi.org/10.7717/peerj.9028#supplemental-information.

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
