# Peer review of "Ancient genetic divergence in bumblebee catfish of the genus Pseudopimelodus (Pseudopimelodidae: Siluriformes) from northwestern South America"

_PeerJ, doi:10.7717/peerj.9028_

## Round 0.1 · original submission · Major Revisions

Dear Authors,

I received two review, and was waiting for an additional review which, unfortunately, never arrived. Overall, I think your MS has merit, and will make a nice study. However, you will need to address comments of the second reviewer, and my comments below regarding the data.

I also agree with the second reviewer that you should produce a species tree.

Discussion should be enriched. You have nice results so explore and discuss them.

The MS needs to be checked for English grammar structure. There are a number of awkward sections, and clear transliterations from Spanish to English.

Specific issues:

COX1 – stop codon in Cruciglanis 2, at position 193 should not be A
RAG2 – missing nucleotide Pimelodus H21 at position 480
misallignment at position 479-480
no reason to have insertions within the RAG2 protein coding region.

Figure 1 - choice of colors terrible, high percentage of men are dautonic and will not be able to differentiate between the subtle shades of green and red. Are Meta and upper Caura the same color (population)?
Figure 1 – indicate what is the Orinoco and Amazon basin
Figure 3 – why is the font red?
Correspondence of colors between Figures 1, 2, 3 and 4 is not clear. I suggest that colors in Figure 1 should correspond to cox1 lineages and then these colors should match Figures 2, 3 and 4.

I look forward to receiving a revised version.

Sincerely,

Tomas Hrbek

Reviewer 1 ·

Basic reporting

The article is clear and unambiguous, with professional English used throughout. The literature references provide sufficient field background/context. The article has a professional structure and the raw data are shared.

Experimental design

The scope of the project is within Aims and Scope of the journal.
The research question well defined, relevant & meaningful. The manuscript states how the research fills an identified knowledge gap.
The research was performed with high technical & ethical standards.
The methods are described with sufficient detail & information to be replicated.

Validity of the findings

The data are robust, and the interpretation are statistically sound. The conclusion is well stated, linked to original research question & limited to supporting results.

Additional comments

An excellent study and worthy of publication in PeerJ. One of the strengths of this study is the use of two fossils to time-calibrate the trees. A drawback is the use of only two genes to construct the molecular phylogeny. Many observers consider a two gene analysis to be deficient in 2018, but in this case the results from each gene are in strong agreement with one another, and make sense in light of the geological literature of the area under investigation. The authors should be advised however that the days of publishing results from just two genes are coming to an end. However I would also like to see an analysis using both genes together, or least an explanation in the text as to why that was not done.

The figures are well-rendered and clear. However it would help the reader if you could include in a figure or figure caption the river basins associated with each rag2 cluster and cox1 lineage. Also, given the non-monophyly of P. bufonis and P. schulzti on the trees, it would be useful to know the location (and river basin) of the type locality of P. schulzti.

The 7.56 Mya date cited (L259) for the uplift of Western Cordillera is unsupported; this precise date is not present in Duque-Caro (1990) or Albert et al (2006), and as far as I know there are no precisely dated estimates for the onset of this uplift. Kellogg and Vega (1995) refer to this uplift as occurring in the "Late Miocene to Pliocene".

·

Basic reporting

The present study tests the hypothesis that the species nominally recognized as Pseudopimelodus bufonius and P. schultzi from Colombia correspond to more than two different evolutionary lineages using two molecular markers (COI – mitochondrial - and RAG2 - nuclear). Although this hypothesis is very common in DNA barcoding studies, the authors did not mention the methodology, what it was not expected for me. But, I think it was interesting use tools from phylogeography field to identify the different lineages and what historical processes were responsible for their diversification.

Overall, the manuscript is well-writing and the language is appropriated (I listed some mistakes in the General comments section). About the literature, I think the authors used classical papers to build the introduction and discuss the results. However, 77.1% of the references have more than five years. There are a lot of new studies using fish as biological model to test the vicariance processes from Northwest of South America. Beyond that, in Brazil, there are studies using morphological and molecular data that identified and describe new species every year. So, I think you could explore more recent papers. To help you, I listed some papers that you can you in the discussion section.

All figures and tables presented in this manuscript are adequate. However, all, except by the Figure 1, need to be reviewed. Several mistakes need to be corrected and the Table 1 have to be improved. I listed these issues in the General comments section. The raw data is in accordance with the Data Sharing policy of PeerJ. However, I strongly recommend to the authors made available all sequences in the GenBank database, not only the haplotypes.

Experimental design

The authors well-designed the experiment to test their hypothesis. The samples are well distributed in different localities considered key to identify the main vicariance processes responsible for the lineages diversification: the sites are localized between important mountains and arch. The authors have the licenses necessary to collect the samples. I am just wondering if they have vouchers. If yes, please, it is necessary to provide the museum numbers.

The methods were described with sufficient detail; however, I have few questions and suggestions to improve this section. Please, see General comments.

Validity of the findings

I think the authors would explore more their findings. For example, for me, the main finding is both species P. bufonius and P. schultzi are not monophyletic. The COI is a molecular marker appropriated to delimit species and this result reveals possible taxonomic issues. I suggest the authors to bring more information about the taxonomic status of the species and write specifically about this great result that could impact the conservation of both species.

The data is robust, but I think the author should provide more information: all sequences have to be the own GenBank numbers and all samples need to be georeferenced. The information detailed are essential for other authors replicate the study.

About the discussion of the results, I think the authors misinterpreted the topology pattern obtained by both molecular markers independently. I suggest the authors to build a tree with both markers (species tree) and re-analyze the data. I listed carefully all this issue in the general comments section.

Additional comments

Major concerns

- Line 122: In the rag2 alignment I saw few problems that need to be solved: (i) in the same site (119, 282, 293, and 306) there is a gap in all sequences, so it is necessary to remove it. (ii) in the sites 479 and 480 there is a problem in the alignment. Please, fix it. (iii) Is it expected the presence of gaps (deleted sites) inside the sequence? I think it is not. The rag2 plays an important role in the rearrangement and recombination of genes of immunoglobulin and T cell receptor molecules, any changes in the open reading frame could be results in serious problems for the cell. Please, check again these gaps (see Results section). (iv) What did you do with the missing data?

- About the previous issue, I do not agree with your statement in the line 173. Because I saw gaps in your rag2 alignment (P. schultzi (H2) site 493, P. bufonius (H21) site 479 and P. schultzi (H18) site 592), I used MEGA v5.2 (Tamura et al., 2011), and when I translated the sequences, used the standard genetic code, I verified at least six stop codons. Did you check this information using MEGA?
Tamura, K., Peterson, D., Peterson, N., Stecher, G., Nei, M., & Kumar, S. (2011). MEGA5: molecular evolutionary genetics analysis using maximum likelihood, evolutionary distance,
and maximum parsimony methods. Molecular Biology and Evolution, 28, 2731–2739.

- Line 145: Why did not the authors use a species tree with both molecular markers? I strongly recommend estimating the divergence time using a tree with both molecular markers. The species tree estimated in BEAST does not require the same individuals to be sampled for each gene nor to match individuals from one to next (Drummond & Bockaert, 2015). Because species tree method requires a priori assignment of individuals to species, the authors can consider as putative species the mitochondrial lineages defined by COI.
Drummond, A. J., & Bouckaert, R. R. (2015). Bayesian evolutionary analysis with BEAST. Cambridge University Press.

- The hypothesis tested by the authors is very interesting and they well-designed the experiment to test it. They expected to find more than two different evolutionary lineages for the two nominal species recognized. They corroborated the hypothesis, but, for my surprise the recognized species was not monophyletic and, unfortunately, the authors did not mention about it. I strongly recommend bringing more information about the species (there are taxonomic problems between them?) and a paragraph discussing this finding.

- Line 247: How did you estimated the divergence between the nominal P. bufonius from Orinoco and Amazon river? They are not in the same clade. You can only tell us that the ancestor of the P. bufonius from the Amazon river (lineage 2) separated from the clade composed by lineages 3, 4 and 5 at approximately 11 Mya.

- Line 254: Neither in the rag2 or coi trees it is not possible to see the splitting between cis- and trans-Andean Pseudopimelodus lineages. If all the divergence events were based on vicariance (as the authors discussed), the topology pattern to affirm about this splitting is to find two main clades: one composed of lineages/species from trans-Andean rivers and other composed of lineages/species from cis-Andean basin (Sivasundar et al., 2001, Albert et al., 2006, Ramirez et al., 2017, Ríos et al., 2017, Machado et al., 2018). Clearly, this does not happen in your results.

Albert, J. S., Lovejoy, N. R., & Crampton, W. G. (2006). Miocene tectonism and the separation of cis-and trans-Andean river basins: Evidence from Neotropical fishes. Journal of South American Earth Sciences, 21(1-2), 14-27.
Machado, C. B., Galetti Jr, P. M., & Carnaval, A. C. (2018). Bayesian analyses detect a history of both vicariance and geodispersal in Neotropical freshwater fishes. Journal of Biogeography.
Ramirez, J. L., Birindelli, J. L., & Galetti, P. M. (2017). A new genus of Anostomidae (Ostariophysi: Characiformes): Diversity, phylogeny and biogeography based on cytogenetic, molecular and morphological
data. Molecular Phylogenetics and Evolution, 107, 308–323.
Ríos, N., Bouza, C., Gutiérrez, V., & García, G. (2017). Species complex delimitation and patterns of population structure at different geographic scales in Neotropical silver catfish (Rhamdia: Heptapteridae). Environmental Biology of Fishes, 100(9), 1047-1067.
Sivasundar, A., Bermingham, E., & Ortí, G. (2001). Population structure and biogeography of migratory freshwater fishes (Prochilodus: Characiformes) in major South American rivers. Molecular Ecology, 10(2), 407-417.

Minor concerns
Keywords: I did not see in the PeerJ’ author guidelines a role for keywords. I suggest one more keyword: “Colombia”. Please, order alphabetically all words.

Introduction
- Line 36: This in-text citation is wrong (Author and year). According to your reference list the correct citation is “Eschmeyer & Fong, 2017”.
- Line 38: Please, organize the genera in alphabetic order.
- Line 41: Does the distribution of both species remain controversial or just for Pseudopimelodus bufonius? In lines 45 to 47 you cited the controversial just for one species.
- Line 44: As you did with Pseudopimelodus bufonius, please provide taxonomic authority for Pseudopimelodus schultzi.
- Line 46: Does Atrato and Sinú rivers from the Caribbean basin? If yes, it is necessary to say this information because in the DoNascimiento et al.’s paper the P. bufonius is restricted to Amazon, Orinoco and Caribbean basins.
- Line 58: - Line 152: In the text-citation with two author the names are separated by “&”, not “and”.
- Line 61 The reference “Lundberg et al., 1998” is wrong. According to your reference list, there is one author in this study. The problem also happens in the line 250.
- Line 66: It is not necessary italicize the abbreviation of gene’ name.
- Line 68: The sentence started in the past “we used”, so in this line, should be “we analyzed”, not “we analyze”.
- Line 68: The right name of cox1 gene is “Cytochrome C oxidase I”. The same issue also happens in the line 112. Please, fix it.

Material and Methods
- Line 85: If the authors used muscle issues I suppose they sacrificed the animals. If I am right, where is the voucher numbers?
- Line 110: I checked the Hardman’s paper to understand the reasons about you choose the primers that amplified a short region of rag2 gene and I did not find the primers rag2-MHF1/rag2-MHR1, but just the primers IcRAG2-F1/IcRAG2-R1. Where did you get those primers?
- Lines 114-117: Is the PCR’s condition correct? The authors performed the PCR in a total volume of 30 µl; however, they use 40.6µl only of sterile nuclease free water. The total volume is much high than 30µl. Can you check the information, please?
- Lines 117-119: After the 32 cycles, do you have a final extension in the thermal profile for rag2? If so, you must inform us.
- Which purification protocol did you use?
- Lines 122 and 143: Please, use the appropriate reference for the software Geneious R10 and PopArt. See links below:
https://support.geneious.com/hc/en-us/articles/227534768-How-do-I-cite-Geneious-in-a-paper-
http://popart.otago.ac.nz/howtocite.shtml
- Line 130: According to DNAsp v5. tutorial, the haplotype diversity measure is represented by “Hd” and not “h”. Please, use the correct form to abbreviate this measure.
- Line 131: Please, insert a dot and comma after “et al.”. The correct form to represent the in-text citation is “Roza et al., 2017”.
- Line 132: It is important to say if the phylogenetic analysis conducted using MrBayes was conducted for each gene or concatenated dataset. When I saw your results, I identified trees for each gene. So, I think is necessary to say this information. My suggestion is: “Phylogenetic analysis for each gene using Bayesian inference was conducted using MrBayes…”.
- Line 152: In the text-citation with two author the names are separated by “&”, not “and”.
- Line 154: According to the BEAST manual (http://phyloworks.org/workshops/DivTime_BEAST2_tutorial_FBD.pdf), the normal distribution is not always appropriate for calibrating a node using fossil information. Can you explain I did you use this distribution to calibrate this node?

Results
Line 172: I am curious: why did you obtain short sequences for these molecular markers? Oliveira et al. (2011) after the alignment obtained 1034 bp for rag2. Ward et al. (2005) using COI universal primers get 655 pb. Did you try to use these primers?
Oliveira, C., Avelino, G. S., Abe, K. T., Mariguela, T. C., Benine, R. C., Ortí, G., ... & e Castro, R. M. C. (2011). Phylogenetic relationships within the speciose family Characidae (Teleostei: Ostariophysi: Characiformes) based on multilocus analysis and extensive ingroup sampling. BMC Evolutionary Biology, 11(1), 275.
Ward, R. D., Zemlak, T. S., Innes, B. H., Last, P. R., & Hebert, P. D. (2005). DNA barcoding Australia's fish species. Philosophical Transactions of the Royal Society of London B: Biological Sciences, 360(1462), 1847-1857.
- Line 183: I am not familiar with de designation “G4” in the substitution models. Can you explain what is it mean?
- Line 185: The posterior probability support for Pseudopimelodus clade recovered by the rag2 was 0.97, not 0.96.
- Lines 205-210: This section was a little confuse for me. In the line 206, do you talk in about the cluster obtained by rag2? If yes, please use cluster than lineage. In the line 207, lineage 2 correspond to the cluster 2? If no, please remove the divergence time of rag2.
- Once, I strongly recommend using the species tree result for this section. It is informative to build a tree with both genes than show each one. I also suggestion to show the 95% HPD values than the absolute ages. For example: 7.9 Mya (95% HPD 4.05 – 11.45 Mya).

Discussion and Conclusions
- Line 256: The reference “Hoorn, 1995” is wrong. There is more than one author in this study. The same issue also happens in the line 250.
- Line 261: Please, insert a dot and comma after et al. The correct form to represent the in-text citation is “Albert et al., 2017”.
- Line 261: Should be “Duque-Caro”, not “Duque-caro”.
- Line 270: And about the species present in the trans-Andean rivers, do you think that correspond to P. bufonius sensu stricto?
- Line 280: I suggest: “at least five different evolutionary lineages revealed by the mtDNA analysis. The interesting result is that neither one corresponds to the two recognized species. All mtDNA lineages exhibit phylogenetic histories concordant with geological history of the Northwestern South America”. The latter sentence is not true. The authors need to check the comments above in the discussion section.
- Line 285: Please, remove “cis and”.
Reference
The reference section and the in-text citations need to be reviewed. Please, use a reference management and follow the PeerJ’s author guidelines.
- In text-citation:
Please, look at the role to use “et al.” (not “et al”, as you did in several times, for example line 72).
Line 36: In-text citations, please, insert a comma after the author’s name.
According to the PeerJ’s author guidelines “Multiple references to the same item should be separated with a semicolon (;) and ordered chronologically”. This latter is an issue in the entire manuscript. Please, correct it.
- Reference list
(i) All references (100%) in this section are not following the author guidelines. Please, correct them.
(ii) There are seven missing references, as following: “Lundberg & Dahdul, 2008”, “Sullivan et al. 2013”, “Hardman, 2006”, “Bandelt et al., 1999”, “Van der Hammen, 1960”, “Albert & Reis, 2011” and “Carvalho et al., 2011”. Please, provide or remove them.
(iii) Do not underline the doi numbers;
(iv) The full title of the Journal need to be italicized. Please, also pay attention in the Journal’ names. I identified names misspelling (line 294, for example).
(v) According to the International Code of Zoological Nomenclature, the genus must be italicized (line 330, for example).

Table and figures
Table 1
- Please, standardize the in-text citation references in the legend. Each reference, include the personal communication, shows incongruence with the author guidelines.
- In the column “GenBank accession number”, the world accession is misspelling.
- The GenBank numbers were obtained for each haplotype. I strongly recommend to the authors deposited all sequences, not haplotypes, in the GenBank. The dataset must be open.
- All samples must be georeferenced. The columns “Country” and “Origin” are not sufficiently. Please, provide the GPS for all samples.

Figure 2
Because the table and figures need to be self-explanatory, I think it is interesting to describe each cluster of Pseudopimelodus as you did in the result section. You can do this in the legend or in a footnote. The same suggestion can be applied for the figures 3, 4 and 5.

Figure 4
- The authors forgotten to mention the blue circle (Magdalena river – middle section).

Figures 5 and 6
I suggest plotting black circles (for example) in the nodes calibrated with fossil.

In the figures 2 to 6, I suggest to the author check again the haplotype numbers. For example, in the figure 3, the lineage 2 (part of nominal species P. bufonius) is composed of the haplotypes 28, 29, 32, 33 and 3. But, according to the table 1, the haplotype 3 is the species P. schultzi. Other issue happens with the figure 3 and 4: the lineages 2 and 3 does not show the same haplotypes.

---

## Round 0.2 · Minor Revisions

Dear Authors,

I have now received one re-review. I agree with the reviewer that your MS has been substantially improved, and I complement you taking the time to revise the figures, and you do a good job discussing your results.
There are few small things that the reviewer would like you to look at, and one you do that, I would be happy to accept your MS for publication.
I look forward to receiving a revised version.
Sincerely,

Tomas Hrbek

·

Basic reporting

This represent the second revision of the manuscript "Ancient genetic divergence in bumblebee catfish of the genus Pseudopimelodus (Pseudopimelodidae: Siluriformes) from northwestern South America" by Rangel-Medrano et al. The authors revised the content of the manuscript and accepted almost all suggestions tha I did. The revised version is much better than the previous, because now the analyses are robust (they added species delimitation analysis and used both markers in the species tree). However, there are several issues that need be corrected, specially the main idea to construct the result section.

Experimental design

See "General comments for the authors".

Validity of the findings

See "General comments for the authors".

Additional comments

New comments:

Material and methods
I propose to change the topic order. It is interesting starts with Genetic diversity and Species delimitation (because you first need to define how will be the putative species), following by Species Tree and phylogenetic relationship and ends with Divergence time estimation topics.

Lines 118-121: To improve the logic, I suggest start with COI marker following by the rag2 marker.

Lines 123-125: The units are not following the rules of International System Units. Please, correct this issue. For example: Should be pmol, not Pmol.

Line 124: Please, spell correct the magnesium chloride formula: It is necessary subscript the number two.

Line 138: Should be “Genetic diversity”, not “Sequence diversity”.

Line 142: The letter “V” indicates the program version should be used in lowercase. This issue also happens in the lines 156, 171, 174, 176. Please, check all the manuscript and correct it.

Line 148: Why did you compare rag2 and cox1 sequences with other species? I did not understand you. Or did you meaning “combine your dataset with sequence present in GenBank database”?

Lines 149-150: It is not possible to understand the following sentence: “Samples were compared with rag2 and cox1 sequences of species in all other genera of the family Pseudopimelodidae and deposited in GenBank using sequences of Pseudoplatystoma magdaleniatum and Pimelodus yuma as outgroup”. How did you submit the sequences in GenBank using other sequences? My suggestion is, if I understood your main idea is:
“We combined our dataset with other species from Pseudopimolididae (Table 1). Sequences from Pseudoplatystoma magdaleniatum and Pimelodus yuma were also add to the final dataset as outgroup. All sequences obtained in this study was submit to GenBank and the accession numbers are in Table 1”.

Line 188: Should be “BEAST v1.8.464”, not “Beast V1.8.464”.

Results
I am a little concern about your results. I think the way how to present them is not the best. My suggestion is a little different:
- You have a lot of figures in this section, a think it will be better if the tree markers stay in the Supplementary section.
- Show first the species delimitation results. Here it is possible to see the hidden diversity in this two nominal species.
- Use a tree with concatened markers to show the phylogenetic relationship among the five lineages (the same determined with the analysis above).
- Show the species tree and divergence time analysis (in the species tree, not and individual tree).
Line 225: Please, standardize the word “dataset”. In the line 218 you wrote “dataset” and in the line you used “data set”.

Line 230: Italicize the nuclear marker name.

Line 230: The Species delimitation analysis also showed five lineages. The authors determined the lineage in this section based on clades observed in tree (subjective analysis). I think you can base your inference in the species delimitation analysis not in the phylogenetic analysis. For this reason, it is important to change the order for this section.

Line 232: Should be “(Figs. 2 and 3, respectively).

Line 254: Insert a space to start a new section.

Lines 259-260: The authors showed the GMYC and bGMYC results, but I did not know if the likelihood of GMYC model is greater than the likelihood of null model and if the p-value is significance. It is necessary to show this information to the readers.

Discussion
Line 313: What is meaning “a.s.l.”

Line 350: Did you get the type locality for P. schultzi, Cereté – Colombia (Dahl, 1955)? If yes, the following sentence is not correct: “indicating at least five new candidate species”, because one P. schulzi lineage represents the nominal species, and the other two can be a potential new candidate species.

Figure 1: There is a color mistake. The circle represent the Magdalena River upper sector is not dark blue in the map. It is a light blue as Cauca River-upper and middle sector. It is not possible to distinguish them.

Figure 6: The Lineage 4 is not a monophyletic in the divergence time estimation. However, in the figure 4, use the same markers it is. Why did this happen?

Figure 6: There is no black circle denoted the calibrated nodes.

Figure 6: The authors missed to insert the node supports on the tree.

Old comments (Answers for the previous review):

- Voucher numbers: I do not agree with the authors answer about this topic. It is not adequate to give the information about the voucher numbers in other paper which was not published yet. The readers need to know this information in the present study. So, I strongly recommend writing this information in the Table 1 or supplement material.

- The author given the information about GPS for all samples as suggest. However, they still use haplotype information in the Table from Supplementary material. What is the problem with that? There are haplotypes (H18, H8, H12, H1, and H19) present in more than one sub-basin and the locality information is wrong. For example: in the supplementary material the haplotype H18 is from “Magdalena River-upper sector in Neiva”, however in the haplotype network is possible to see that H18 is also present in Cauca River lower sector and San Jorge River. I strongly recommend giving the information based on individual than haplotype.

---

## Round 0.3 · accepted · Accept

Dear authors,

I have received and evaluated your manuscript. I am happy to inform you that I am happy with the revisions you made, and the clarity and objectivity you brought into you analyses and writing. I also appreciate the very nice contribution you made to our knowledge of aquatic biodiversity of Colombia, and the recognition of the importance of considering evolutionary lineages in conservation irrespective whether they have been formally described as species or not.

Congratulations on a job well done.

Sincerely,

Tomas Hrbek

P.S. When correcting your page proofs, please note that on line 161 you want to cite STARTBEAST2 and not *BEAST.